# High-Intensity Interval Training Induces Protein Lactylation in Different Tissues of Mice with Specificity and Time Dependence

**DOI:** 10.3390/metabo13050647

**Published:** 2023-05-09

**Authors:** Wenhua Huang, Jie Su, Xuefei Chen, Yanjun Li, Zheng Xing, Lanlan Guo, Shitian Li, Jing Zhang

**Affiliations:** 1School of P.E. and Sports Science, Beijing Normal University, Beijing 100875, China; 2Department of Physical Education, University of International Business and Economics, Beijing 100029, China

**Keywords:** high-intensity interval training, lysine lactylation, energy metabolism, proteomics

## Abstract

Protein lysine lactylation (Kla) is a novel protein acylation reported in recent years, which plays an important role in the development of several diseases with pathologically elevated lactate levels, such as tumors. The concentration of lactate as a donor is directly related to the Kla level. High-intensity interval training (HIIT) is a workout pattern that has positive effects in many metabolic diseases, but the mechanisms by which HIIT promotes health are not yet clear. Lactate is the main metabolite of HIIT, and it is unknown as to whether high lactate during HIIT can induce changes in Kla levels, as well as whether Kla levels differ in different tissues and how time-dependent Kla levels are. In this study, we observed the specificity and time-dependent effects of a single HIIT on the regulation of Kla in mouse tissues. In addition, we aimed to select tissues with high Kla specificity and obvious time dependence for lactylation quantitative omics and analyze the possible biological targets of HIIT-induced Kla regulation. A single HIIT induces Kla in tissues with high lactate uptake and metabolism, such as iWAT, BAT, soleus muscle and liver proteins, and Kla levels peak at 24 h after HIIT and return to steady state at 72 h. Kla proteins in iWAT may affect pathways related to glycolipid metabolism and are highly associated with de novo synthesis. It is speculated that the changes in energy expenditure, lipolytic effects and metabolic characteristics during the recovery period after HIIT may be related to the regulation of Kla in iWAT.

## 1. Introduction

Protein lysine lactylation (Kla) is a novel protein acylation reported in *Nature* in 2019 [1] which refers to the state in which lactyl-coenzyme A is covalently bound to lysine residues of proteins catalyzed by lysine acyltransferases. The concentration of lactate as a donor precursor is directly related to the level of lactylation, and existing studies have shown that lactylation plays an important role in the development of several diseases with pathologically elevated lactate levels, such as tumors and sepsis. Lactate accumulation in the tumor microenvironment promotes immunosuppression by inducing histone lactylation in tumor-infiltrating myeloid cells, thus upregulating methyltransferase-like 3 [2]; additionally, lactate-induced histone lactylation in melanoma promotes YTH structural domain family protein 2 expression, thus driving tumorigenesis [3]. Moreover, lactate promotes lactylation of high-mobility group box-1 in macrophages in sepsis and affects their survival [4]. In addition, elevated lactate levels in the nervous system are also involved in neuroexcitation and anxiety-induced behavioral abnormalities in brain cells, as well as Alzheimer’s disease-induced dysfunction, through protein lactylation [5,6]. Zhang et al. found that, in isolated cultured human HeLa cells and mouse bone marrow-derived macrophages, accumulated lactate following enhanced glycolysis by rotenone treatment similarly induced protein lactylation. These studies suggest that elevated lactate levels in vivo may act biologically through a novel mechanism of protein lactylation [1]. However, the effect of elevated lactate levels that are induced during exercise on protein lactylation in the body has not been reported, and its biological role is currently unknown.

High-intensity interval training (HIIT) is a form of exercise in which short periods of high-intensity exercise and low-intensity exercise/rest are alternated and repeated several times [7]. Numerous studies have confirmed the positive effects of HIIT on fat loss, cardiovascular disease, diabetes, sarcopenia and nonalcoholic fatty liver disease [8,9,10]; however, the mechanism of action of HIIT for health promotion is not yet clear. Lactate is the main metabolite of HIIT, and it is currently unknown as to whether repeated and multiple exposures of the body to high lactate during HIIT can induce changes in lactylation levels, as well as whether lactylation levels differ in different tissues and how time-dependent lactylation levels are.

A large number of studies have shown that exercise modulates protein acylation as an important mechanism for health improvement, with some studies reporting that exercise is involved in the promotion of cognitive function in the nervous system by regulating histone acylation in the hippocampus [11]. In addition, exercise also influences exercise effects and metabolic health by modulating the acylations of various proteases in skeletal muscle [12]. The role of altered lactylation induced by high-intensity interval training in health is of interest [13].

In this study, we aimed to perform a whole-protein pan-Kla-level assay in HIIT mice and collect tissues of cardiac muscle, skeletal muscle, white fat, brown fat, liver and brain at multiple time points during rest and after exercise to observe the specificity and time-dependent effects of single HIIT on the regulation of protein lactylation in body tissues. In addition, we aimed to select tissues with high Kla specificity and obvious time-dependent effects for lactylation quantitative omics, analyze the possible biological targets of HIIT-induced Kla regulation and explore the Kla mechanism of HIIT for health promotion.

## 2. Materials and Methods

### 2.1. Experimental Animal Grouping

Eight-week-old male C57/BL6 mice weighing 23.23 ± 1.09 g were purchased from Beijing Vital River Laboratory Animal Technology Co., Ltd. (Beijing, China), experimental animal production license No. SCXK (Beijing) 2016-0006. Mice were randomly divided into the control group (Con) and the HIIT exercise group (HIIT), which were further divided into 0/2/24/48/72 h postexercise groups according to the different time points of sacrifice at the end of HIIT exercise. The mice were housed in separate cages at a room temperature of 22 ± 5 °C, humidity of 50 ± 10%, alternating light and dark cycles of 12 h and free access to food and water. All of the feedings were provided by Huafukang Animal Experimental Feed Factory (Beijing, China).

### 2.2. Exercise Protocol

Mice were first subjected to adaptive running platform exercise, and maximum running speed was determined by increasing the treadmill speed by 1 m/min every 2 min until the mice were unable to keep up with the treadmill speed for more than 10 consecutive seconds under light stimulation with a wooden stick to determine the maximum running speed (Smax) [14]. The following HIIT exercise protocol was designed according to the maximum running speed: 40% Smax warm-up for 5 min, followed by 85% Smax exercise for 1.5 min plus 45% Smax exercise for 2 min in one set, repeated for 9 sets and ending with 40% Smax relaxation for 3 min.

The mice were subjected to HIIT exercise according to the protocol, and the mice were sacrificed immediately and at 2, 24, 48 and 72 h after the end of exercise.

### 2.3. Blood Lactate Assay in Mice

At different time points after exercise, blood was collected from the tail vein of mice for blood lactate determination, and blood lactate concentrations were measured by using blood lactate test strips (EKF, Lactate-scout, Leipzig, Germany) and a lactate tester (EKF, Lactate-scout, Leipzig, Germany).

### 2.4. Detection of Energy Expenditure in Mice

Energy expenditure in mice was measured by using the OxyletPro metabolic assay system (Panlab, Barcelona, Spain). The following procedures were performed: the mice were placed in a respiratory metabolic cage for 24 h to acclimatize before HIIT exercise, followed by the HIIT running exercise; immediately after the exercise, the mice were placed back into the respiratory metabolic cage and continuously monitored for 24 h to measure the data of energy consumption, oxygen consumption, activity and food intake after which their total energy consumption was calculated.

### 2.5. Protein Extraction and Western Blotting

The inguinal white adipose tissue (iWAT), epididymal white adipose tissue (eWAT), brown adipose tissue (BAT), liver, heart, brain, soleus and gastrocnemius muscle tissues were collected from C57/BL6 mice. Tissues were homogenized in lysis buffer, and total protein was quantified. Protein solutions were separated via electrophoresis by using 10–12% Bis-Tris gels and transferred to polyvinylidene difluoride (PVDF) membranes. The membranes were incubated with 5% skim milk for 2 h and then incubated overnight at 4 °C with primary antibody (anti-L-lactyl lysine rabbit mAb, 1:1000, PTM-1401, Jingjie PTM BioLab Co. Ltd., Hangzhou, China; GAPDH, 1:8000, CST5174, Cell Signaling Technology, Boston, USA; β-actin, 1:4000, E030110, Earthox, San Juan Island, WA, USA) diluted in TBST, after which they were finally incubated for 1 h with secondary antibody (goat anti-rabbit IgG, 1:8000, Earthox, San Juan Island, WA, USA) diluted with TBST. Membranes were then washed three times with PBS-T. Chemiluminescence reagent (WBKLS0100, Millipore, Burlington, NJ, USA) and a luminescence imaging analyzer (Amersham Imager 680, General Electric, Boston, MA, USA) were used for protein band detection. The relative protein expression was analyzed based on the gray value for GAPDH or β-actin.

### 2.6. Four-Dimensional Label-Free Lactylation Quantitative Proteomics and Analysis

Four-dimensional label-free lactylation quantitative proteomics in mouse iWAT was performed by Jingjie PTM BioLab Co., Ltd. (Hangzhou, China).

#### 2.6.1. Protein Extraction and Trypsin Digestion

Samples were removed from −80 °C and then fully ground to powder with liquid nitrogen. The samples in each group were added to 4 times the volume of extraction buffer (containing 10 mM DL-dithiothreitol, 1% protease inhibitor cocktail, 3 μM TSA and 50 mM NAM) for ultrasonic lysis. An equal volume of Tris-balanced phenol was added, and the samples were centrifuged at 5500× *g* at 4 °C for 10 min. The supernatant was collected, and 5 times the volume of 0.1 M ammonium acetate/methanol was added for precipitation overnight. The precipitate was successively washed with methanol followed by acetone. Finally, 8 M urea was used for redissolution, and the protein concentration was determined by using a BCA kit.

The protein in each sample was enzymatically hydrolyzed in equal quantities, and the volume was adjusted to be consistent with the lysate. Subsequently, 20% TCA was slowly added, and the sample was vortexed and allowed to precipitate at 4 °C for 2 h. Centrifugation was performed at 4500× *g* for 5 min, and the precipitate was washed with precooled acetone 2–3 times. After drying off the precipitate, 200 mM TEAB was added. Afterwards, trypsin was added at a ratio of 1:50 (protease: protein, M/M), and the sample was digested overnight. Dithiothreitol (DTT) was added to a final concentration of 5 mM, and the sample was reduced at 56 °C for 30 min. Iodine acetamide (IAA) was subsequently added to a final concentration of 11 mM, and the mixture was incubated for 15 min at room temperature in the absence of light.

#### 2.6.2. Pan-Kla Antibody-Based PTM Enrichment

Peptides were dissolved in IP buffer solution (100 mM NaCl, 1 mM EDTA, 50 mM Tris-HCl, 0.5% NP-40, pH 8.0), and the supernatant was transferred to prewashed lactylation resin (antibody resin Art. 5798717296493442124, Hangzhou Jingjie Biotechnology Co., Ltd., PTM Bio, Hangzhou, China), placed on a rotating shaker at 4 °C and gently shaken overnight. After incubation, the resin was washed 4 times with IP buffer solution and twice with deionized water. Finally, 0.1% trifluoroacetic acid was used to elute the resin-bound peptide three times. The eluent was collected and vacuum dried. After being drained, the salt was removed according to the C18 ZipTips instructions (ZTC18S, Millipore, Burlington, NJ, USA); after being vacuum frozen and drained, the liquid was obtained for mass analysis.

#### 2.6.3. Quantitative Proteomic Analysis via LC–MS/MS

The peptides were dissolved in buffer A and separated via ultra-performance liquid chromatography by using a nanoElute system. Buffer A was 0.1% formic acid and 2% acetonitrile in water. Buffer B was 0.1% formic acid in acetonitrile solution. The liquid phase gradient settings were as follows: 0–44 min, 6–22% B; 44–56 min, 22–30% B; 56–58 min, 30–80% B; 58–60 min, 80% B. The flow rate was maintained at 300 nL/min. Moreover, the peptides were separated, injected into the capillary ion source and analyzed by using a timsTOF Pro mass spectrometer.

#### 2.6.4. Bioinformatics Analysis

Gene Ontology (GO) annotation of the proteome was performed with eggnog-mapper software (version 2.0). Identified protein domain functional descriptions were annotated by using InterProScan (version 5.14-53.0) based on the protein sequence alignment method, and the InterPro domain database was used. The Kyoto Encyclopedia of Genes and Genomes (KEGG) database was used to annotate the protein pathways. These pathways were classified into hierarchical categories according to the KEGG website. All of the differentially expressed protein database accessions or sequences were searched against the STRING database (version 11.0) for protein–protein interactions. WoLF PSORT (version 0.2), which is subcellular localization prediction software, was used to predict subcellular localization. Soft MoMo (version 5.0.2) was used to analyze the model of sequences comprising amino acids in specific positions of modify-21-mers in all of the protein sequences.

### 2.7. Data Analyses and Statistics

In this study, data are presented as the mean ± SD. Statistical analyses were performed by using one-way ANOVA followed by the Newman–Keuls posttest. A *p* value < 0.05 was considered to be statistically significant. Prism (Version 9.0) was used to verify the normality of the data. Moreover, the cross-sectional area of adipocytes was calculated via ImageJ (Version 1.53a.). The grayscale values of the protein bands were calculated by using Image Processing and Analysis in Java (ImageJ, version 1.53a).

## 3. Results

### 3.1. Changes in Blood Lactate and Energy Expenditure after HIIT

The detailed exercise protocol and time points of sampling are shown in Figure 1A. Blood lactate concentrations after HIIT demonstrated a significant increase in blood lactate concentrations immediately after exercise (*p* < 0.01); blood lactate concentrations then slowly decreased and remained higher than in the Con group 5 min after HIIT (*p* < 0.05). Furthermore, blood lactate concentrations were not significantly different from the Con group after 15, 30 and 60 min (Figure 1B).

Changes in energy metabolism after HIIT showed that the energy expenditure of the HIIT group was higher than that of the Con group 24 h after HIIT exercise in mice, which occurred both during the daytime and nighttime (Figure 1C); in addition, the total energy expenditure of the HIIT group was significantly higher than that of the Con group immediately after exercise, as well as at 2 h after exercise and 24 h after exercise (*p* < 0.01, Figure 1D).

### 3.2. Effect of HIIT on Lactylation of Proteins from Different Tissues

#### 3.2.1. Effect of HIIT on Lactylation of Adipose Tissue Proteins

The Western blotting results showed that Kla levels were significantly higher in the 24 h group than in the Con group in iWAT (*p* < 0.05, Figure 2A); in BAT, Kla levels were significantly higher in the 0 h, 2 h and 24 h groups than in the Con group (0 h and 2 h: *p* < 0.05; 24 h: *p* < 0.01, Figure 2B). Moreover, in eWAT, the time points were not significantly different (Figure 2C). This suggests that, in adipose tissue, Kla levels in iWAT and BAT appear to be idiosyncratically upregulated after HIIT.

#### 3.2.2. Effect of HIIT on Lactylation of Cardiac and Skeletal Muscle Proteins

Western blotting results showed that Kla levels were significantly higher in the 24 h group than in the Con group and at 0 h in the soleus muscle (*p* < 0.01, Figure 3A); additionally, there were no significant changes in Kla levels at any time point in the cardiac and gastrocnemius muscles (Figure 3B,C). This indicates that there was a specific upregulation of Kla levels after HIIT in the soleus muscle.

#### 3.2.3. Effect of HIIT on Lactylation of Other Tissue Proteins

Western blotting results showed that Kla levels were higher in the 24 h group than in the Con group in the liver (*p* < 0.05, Figure 4A); moreover, there was no significant change in Kla levels at any time point in the brain (Figure 4B). This indicates that there was a specific upregulation of Kla levels in the liver after HIIT.

#### 3.2.4. Time-Dependent Effect of HIIT on Tissues with Specific Changes in Lactylation

The Western blotting results showed that iWAT, BAT, soleus muscle and liver had significantly increased Kla levels within 24 h after HIIT. To explore the peaks and trends in Kla levels after HIIT, these tissues were tested for changes in Kla levels at 48 and 72 h after HIIT. The results showed that iWAT had significantly lower Kla levels at 48 and 72 h than at 24 h (*p* < 0.05, Figure 5A); additionally, BAT had significantly lower Kla levels at 72 h than at 24 h (*p* < 0.01, Figure 5B). Soleus muscle Kla levels at 48 and 72 h remained significantly higher than that in the Con group (*p* < 0.05, Figure 5C); moreover, liver Kla levels at 72 h were significantly lower than those at 24 h (*p* < 0.05, Figure 5D). It was demonstrated that Kla levels in these tissues after HIIT peaked at 24 h and then declined in a stepwise fashion, with the vast majority of levels returning to steady state levels at 72 h.

### 3.3. Quantitative Proteomic Analysis of Lysine Lactylation in Mouse iWAT

#### 3.3.1. Kla-Upregulated Protein Screening

The iWAT lysine lactylation of quantitative proteomic results showed that there were 25 proteins and 37 sites of Kla modification that were upregulated in the HIIT group compared with the Con group (Figure 6A); detailed Kla upregulation site volcanoes and heatmaps of upregulated proteins are shown in Figure 6B,C. Among them, fatty acid synthetase (Fasn) had eight Kla upregulation sites, ATP-citrate lyase (Acly) had four upregulated sites, superoxide dismutase 1 (SOD1) and pyruvate carboxylase (PC) had two upregulated sites and all others had one upregulated site (Figure 6D).

#### 3.3.2. Functional Classification of Kla-Upregulated Proteins

After classifying Kla-upregulated proteins according to GO and subcellular localization, the results of biological processes in GO showed that the vast majority of Kla-upregulated proteins are involved in metabolic and cellular processes; additionally, cellular components demonstrated that they are involved in cellular and extracellular compositions, and the molecular functions showed that they are overwhelmingly involved in adhesive and catalytic activities (Figure 7A). The subcellular localization classification showed that the vast majority of Kla-upregulated proteins were located in the cytoplasm (56%), followed by mitochondria (28%), extracellular space (12%) and nucleus (4%, Figure 7B). It is suggested that the screened differentially upregulated Kla proteins in iWAT (likely located in the cytoplasm and mitochondria) are involved in cellular metabolic processes, which are possibly regulated by molecular functions of complexes and catalytic activity.

#### 3.3.3. Kla Upregulates Protein GO Enrichment Function

For the annotation of all of the identified proteins and the screening of differentially upregulated proteins, we performed GO classification enrichment analysis comparing differentially upregulated proteins in the HIIT and Con groups to determine whether there was a significant trend of enrichment in certain functional types. The *p* values obtained from the enrichment test (Fisher’s exact test) are presented as a bubble plot showing the functional categories and pathways in which differentially expressed proteins are significantly enriched (*p* < 0.05). The results of the top 20 most significantly enriched categories are presented in bubble plots. Additionally, the vertical axis of the bubble plot is the functional classification or pathway, and the horizontal axis value is the log2-transformed value of the fold change in the proportion of differential proteins in that functional type compared to the proportion of identified proteins. Moreover, the circle color indicates the enrichment significance *p* value, and the circle size indicates the number of differential proteins in the functional classification or pathway.

Enrichment analysis of molecular functions showed that proteins involved in hydrolase, ATP, NADP and phosphatase binding were lactylated at higher levels after HIIT (Figure 8A); additionally, enrichment analysis of cellular components showed that proteins involved in secretory vesicles, myelin and pyruvate dehydrogenase complexes were lactylated at higher levels (Figure 8B). Furthermore, enrichment analysis of biological processes showed that proteins were involved in phospho-nucleoside metabolic processes, organic phosphate biosynthesis, coenzyme biosynthesis and dicarboxylic acid metabolism (Figure 8C). From the GO perspective, it is suggested that HIIT-induced lactylation of iWAT proteins may mediate changes in the biological processes of energy metabolism through various enzymes and binding processes related to glycolipid metabolism.

#### 3.3.4. KEGG Enrichment Pathway of Kla-Upregulated Proteins

KEGG is an information network linking known molecular interactions, and the differentially Kla-upregulated protein KEGG pathway that was obtained from the abovementioned enrichment analysis was visualized in the form of a web page, with the differentially upregulated protein indicated in red in the figure. The results showed that differentially Kla-upregulated proteins were mainly enriched in the tricarboxylic acid cycle, pyruvate metabolism, glycolysis/gluconeogenesis and oxidative phosphorylation metabolic pathways (Figure 9). Moreover, KEGG further supported the abovementioned speculation of GO enrichment, and all four enriched pathways were highly related to glycolipid metabolism and energy metabolism, thus suggesting that HIIT-mediated iWAT lactylation proteins may be involved in the regulation of energy metabolism.

## 4. Discussion

Lactate is the most well-known small molecule in the field of sports science; from early time periods, it was referred to as a “metabolic waste”, and in later time periods, it was known as an “energy fuel”. Lactate is also classified as being a metabolic small molecule and signaling molecule. The role of lactate in exercise is again becoming a focus of research [15].

### 4.1. Tissue Specificity of HIIT-Induced Kla

The mice were sacrificed and sampled immediately and 2 h and 24 h after HIIT, and different tissues were examined for protein pan-Kla. To facilitate the quantification of Kla levels, representative bands with significant Kla levels in each tissue were selected for quantitative analysis.

The effects of exercise on protein lactylation in different tissues have not yet been reported. In this study, HIIT was found to induce upregulation of protein lactylation levels in iWAT, BAT, soleus muscle and liver and they peaked at 24 h.

The results of this study suggest that the lactylation that occurs postexercise is tissue-specific, and lactylation mainly occurs in metabolically active tissues such as the soleus muscle, adipose tissue and liver, which also have the characteristics of lactate metabolism and are sites of lactate uptake, removal and utilization. Representative tissues for lactate for oxidative energy supply are the soleus muscle and cardiac muscle; specifically, slow-twitch fibers have a large amount of monocarboxylate transporter 1 (MCT1) that is responsible for transporting lactate into myocytes, as well as little MCT4 responsible for the outward transport of lactate [16]. During exercise, MCT1 mediates the entry of lactate into slow-twitch fibers for direct oxidative energy supply. The healthy liver has a higher net lactate clearance than any other organ, accounting for 70% of systemic clearance [17]. Moreover, lactate uptake by adipose tissue during exercise may be a way to address elevated blood lactate levels and may promote adipose tissue browning [18,19]. In addition to its otherwise known functions in these metabolically active tissues, the present study found that lactate may play other biological roles by promoting protein lactylation.

Numerous previous studies have reported that exercise induces muscle and liver acylation levels and is associated with changes in metabolite levels [20,21,22,23,24]. Our study likewise implies that lactate produced by exercise may induce similar effects.

In the brain, gastrocnemius and eWAT Kla levels did not significantly change after exercise, which is likely due to the fact that these tissues are not the main locations where blood lactate is taken up for metabolism. The presence of the blood-brain barrier makes it difficult for blood lactate to enter the brain; moreover, the very small amount of blood lactate that enters the brain during exercise is also directly oxidized for energy supply [25]. The gastrocnemius muscle is anaerobically glycolytically energized in HIIT, thus producing large amounts of lactate that are transported out of the myocyte via MCT4 and that enters the circulation through the lactate shuttle; additionally, the gastrocnemius muscle has reduced lactate levels and is the most important producer (rather than utilizer) of lactate during exercise [26]. The abovementioned study suggests that the lactate level of the tissue may be responsible for influencing its lactylation level.

In addition to lactate concentration as an influence of Kla donors, both lactyltransferase and deacetylases (HDACs) are important influencing factors; however, there are fewer studies on exercise intervention of acylated metabolizing enzymes, and the relationship between these factors is uncertain. It has been found that lactate also inhibits the activity of protein HDACs [27,28], which are the specific deacylases in Kla [29]. This scenario may also be one of the reasons why lactate produced by HIIT idiosyncratically regulates the increased levels of Kla.

### 4.2. Time Dependence of HIIT-Induced Kla

Exercise induces a differential regulation of the time-dependent nature of different posttranslational modifications (PTMs). A time-dependent effect of exercise on PTMs of skeletal muscle in animals and humans has been reported in previous studies [30,31]. It is evident that PTMs are dynamic modifications that take time to change [32]. However, there are no straightforward studies on the time dependence of changes in the level of lactylation after exercise.

In this study, changes in Kla levels were continuously observed for 72 h after a single HIIT exercise session, and several tissues were found to have a stepwise increase in Kla, wherein it peaked at 24 h and then gradually reduced. This is similar to the results of previous Kla kinetic (24 h) studies; specifically, the activation of bone marrow-derived macrophages (BMDMs) with lipopolysaccharide and interferon-γ resulted in a significant increase in Kla after 16 h, peaking at 24 h [1]. In addition, after the treatment of BMDMs with lactate BMDMs, Kla significantly increased after 24 h [33]. Most of the current studies have examined changes during the 24 h following the induction of Kla; changes in Kla levels after 24 h have not been reported. Our results showed that iWAT, BAT and liver Kla levels quickly decreased, whereas Kla levels in soleus muscle slowly decreased. However, the regulatory mechanism responsible for this effect is unclear.

In this study, we observed that HIIT, in addition to producing large amounts of lactate, increased excess postexercise oxygen consumption and postexercise energy metabolism. After HIIT, the energy expenditure of mice was higher than that of control mice for 24 h, and the total energy metabolism was significantly higher than that of control mice immediately and 2 h and 24 h after HIIT. These results are in good agreement with the time-dependent regulation of tissue protein lactylation. Although the effect of elevated blood lactate by HIIT rapidly disappeared, the effect of enhanced energy metabolism was maintained for more than 24 h. It is hypothesized that HIIT-induced enhancement of lactylation is involved in promoting energy expenditure by regulating the biological functions of multiple proteins.

### 4.3. Biological Effects of HIIT-Induced Kla

One of the characteristics of HIIT in training scenarios is its powerful fat loss effect and postexercise energy depletion effect. In this study, we found that the Kla level of iWAT showed significant changes after HIIT, which increased with time. iWAT is the main tissue affected by HIIT and is closely related to the purpose and effects of exercise. Moreover, iWAT Kla level changes are related to metabolic remodeling, which is a new entry point to elucidate the effect of HIIT; thus, 4D label-free lactylation quantitative omics was used to detect differential proteins and sites of lactylation in iWAT.

As a type of acylation, Kla has many characteristics in common with other acylations. The first commonality is that it occurs on histone and nonhistone proteins and has different effects on function. The vast majority of current reports on lactylation concern histone lactylation and fewer are on nonhistone lactylation. When histones are modified by acylation, they promote gene transcription and expression, whereas nonhistone acylations cause changes in protein structure, which correspondingly alter protein interactions. Furthermore, acylations that occur in metabolic regulatory proteins tend to inhibit enzyme activity and impair metabolic processes.

The classification of subcellular localization by lactylation quantitative proteomics demonstrated that more than 80% of the lactylation proteins in iWAT are localized in the cytoplasm and mitochondria, which are the most active locations for cellular substance metabolism and energy metabolism; in addition, the nonhistone Kla may affect their active functions.

The same GO classification of biological processes demonstrated a high percentage of lactylation proteins involved in phosphonucleoside metabolic processes and organophosphate biosynthesis processes, which are important components of energy metabolism and life processes [34]. Additionally, coenzyme biosynthesis processes and dicarboxylic acid metabolism processes have a higher degree of Kla, which are equally related to metabolic homeostasis, glucose metabolism and amino acid metabolic processes [35,36,37]. The abovementioned GO analysis demonstrated HIIT-induced lactylation of iWAT protein, which seems to be closely related to the process of material energy metabolism.

KEGG-enriched pathways were demonstrated to be mainly enriched in the tricarboxylic acid cycle, pyruvate metabolism, glycolysis/glycogenesis and oxidative phosphorylation metabolic pathways. Several reports have suggested a close relationship between these pathways in fat loss, fat browning and metabolic regulation [38,39,40,41]. The enrichment pathways of Kla-upregulated proteins in KEGG are all highly correlated with glucolipid metabolism and energy, which may be one of the reasons why HIIT regulates lipid metabolism.

In our experiments, we verified that energy expenditure after HIIT was significantly higher in mice than in control mice and that energy expenditure after HIIT mainly originated from lipid metabolism [42]. Additionally, in this study, we found that both iWAT and BAT proteins exhibited increased Kla levels; thus, it is speculated that the Kla of proteins in adipose tissue may be related to their metabolic activity.

The lactylation quantitative proteomics results revealed that lactate produced by HIIT induced Kla of several proteins and sites in iWAT, most notably Fasn and Acly, with eight and four sites of lactylation being observed, respectively. In contrast, Fasn and Acly are the key proteases for de novo lipogenesis [43], and they are the core proteins that control fat metabolism [44]. The acetylation of Fasn and Acly has been found to affect their stability and activity, thus affecting their functions [45,46]. Although Kla is acylated and acetylated [47], it is possible that these proteins produce the same effects after Kla is acylated, which can possibly affect the metabolic process of de novo lipogenesis and mediate a greater rate of HIIT lipolysis than MICT lipolysis. In addition, it has been found that Fasn can be lactylated and affect lipid metabolism in nonalcoholic fatty liver disease [48].

Based on the abovementioned evidence, it can be speculated that the lipid metabolic effects resulting after HIIT may be related to the action of lactate, and one of the mechanisms of this regulation of lipid metabolism may be closely related to the modifications of lipid metabolism-related enzymes.

The present study still has some limitations: (1) the effect of long-term HIIT on lactylation in each tissue could not be investigated in our study, and whether the long-term effect is consistent with single HIIT deserves further confirmation. (2) In addition, this study used only Western blotting as a screen for tissue-specific changes in lactylation after HIIT, which may not be completely accurate. The regulation of lactylation is diverse, both elevated and decreased, and measuring the precise effect of HIIT on lactylation of tissues also requires histology to be combined with lactylation proteomics.

This report investigates which tissues HIIT may affect regarding the specific upregulation of lactylation, providing a direction and basis for further subsequent targeting of lactylation-mediated biological functions and functional validation of target proteins in different tissues; the specificity of the upregulation of tissue lactylation levels after HIIT was also explored, mainly from the perspective of lactate metabolism, and the existence of specific regulation of related metabolic enzymes also deserves further attention; the exact mechanisms accounting for the time-dependent changes in tissue lactylation after HIIT are not yet clear and equally deserve further exploration.

## 5. Conclusions

A single HIIT induces lactylation in tissues with high lactate uptake and metabolism, such as iWAT, BAT, soleus muscle and liver proteins, and Kla levels peak at 24 h after HIIT and return to steady state levels at 72 h. However, the regulatory mechanisms need to be explored in depth. Lactylation proteins in iWAT may affect pathways related to glycolipid metabolism and are highly associated with de novo synthesis and energy metabolism. It is speculated that the changes in energy expenditure, lipolytic effects and metabolic characteristics during the recovery period after HIIT may be related to the regulation of protein lactylation in inguinal white adipose tissue.

## Figures and Tables

**Figure 1 metabolites-13-00647-f001:**
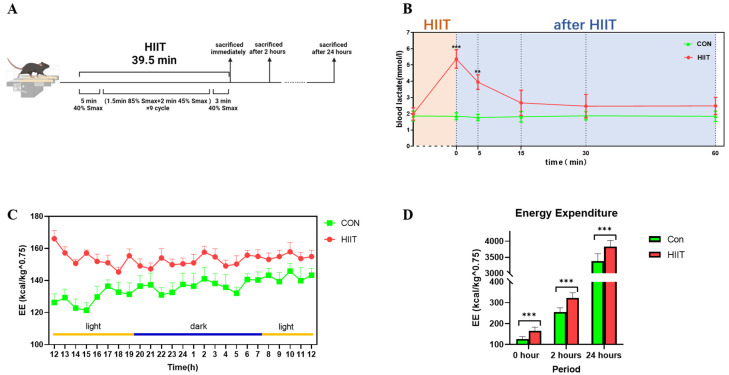
Changes in blood lactate and energy expenditure in mice after HIIT. (**A**) HIIT mouse protocol and sacrifice time; (**B**) blood lactate changes in Con and HIIT mice; (**C**) energy expenditure changes in Con and HIIT mice; and (**D**) energy expenditure in Con and HIIT mice at 0, 2 h and 24 h. ** *p* < 0.01, *** *p* < 0.001.

**Figure 2 metabolites-13-00647-f002:**
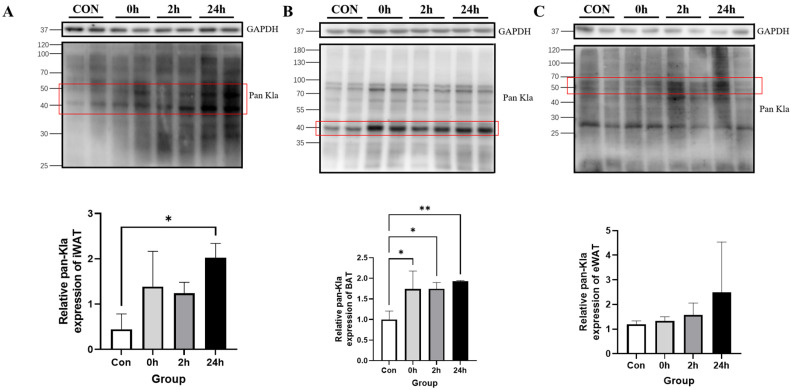
Changes in protein pan Kla in adipose tissue immediately, 2 h and 24 h after HIIT. (**A**) iWAT pan-Kla detection and quantification; (**B**) BAT pan-Kla detection and quantification; and (**C**) eWAT pan-Kla detection and quantification. * *p* < 0.05, ** *p* < 0.01, the red squares are representative bands for statistical analysis.

**Figure 3 metabolites-13-00647-f003:**
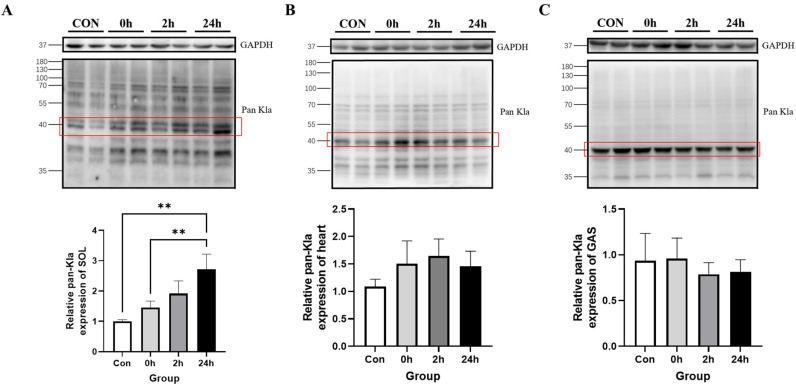
Changes in protein pan-Kla in cardiac and skeletal muscle immediately, 2 h and 24 h after HIIT. (**A**) Soleus muscle pan-Kla detection and its quantification; (**B**) heart pan-Kla detection and its quantification; and (**C**) gastrocnemius muscle pan-Kla detection and its quantification. ** *p* < 0.01, the red squares are representative bands for statistical analysis.

**Figure 4 metabolites-13-00647-f004:**
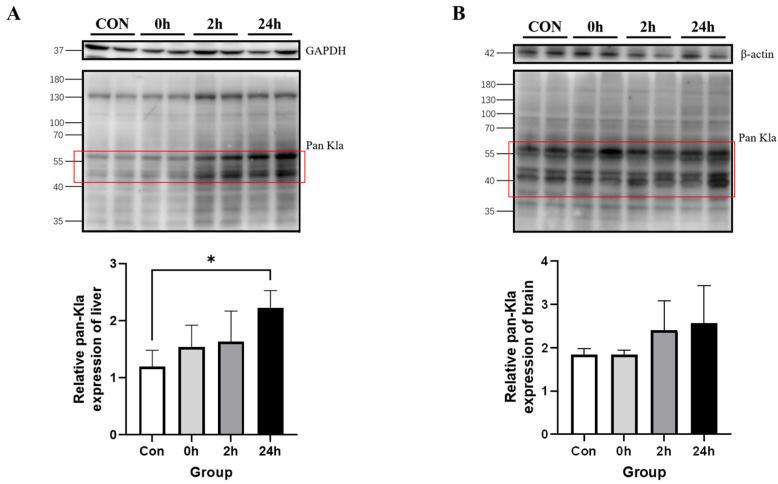
Changes in protein pan-Kla in other tissues immediately, 2 h and 24 h after HIIT. (**A**) Liver pan-Kla detection and quantification; (**B**) brain pan-Kla detection and quantification. * *p* < 0.05, the red squares are representative bands for statistical analysis.

**Figure 5 metabolites-13-00647-f005:**
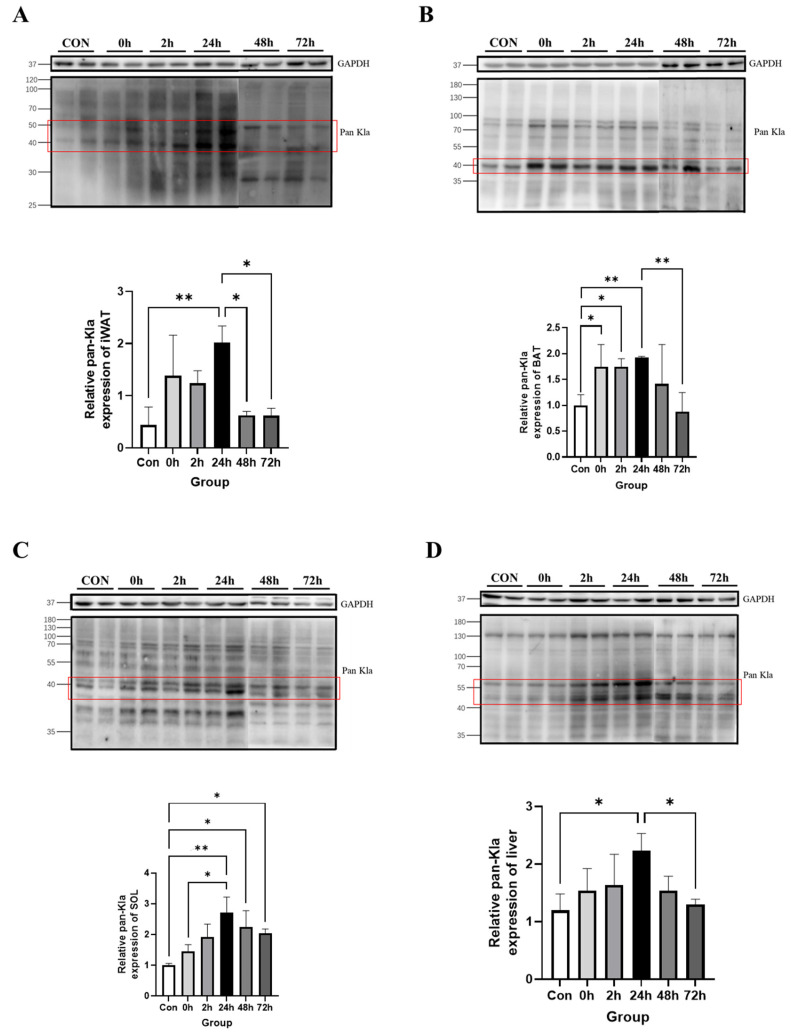
Changes in protein pan-Kla in iWAT, BAT, soleus muscle and liver tissues after HIIT. (**A**) iWAT pan-Kla detection and quantification; (**B**) BAT pan-Kla detection and quantification; (**C**) soleus muscle pan-Kla detection and quantification; and (**D**) liver pan-Kla detection and quantification. * *p* < 0.05, ** *p* < 0.01, the red squares are representative bands for statistical analysis.

**Figure 6 metabolites-13-00647-f006:**
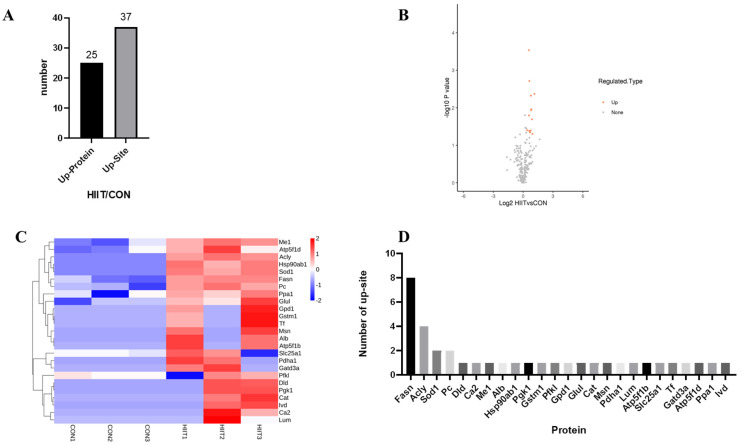
Screening of differentially Kla-upregulated proteins. (**A**) The number of iWAT Kla-upregulated proteins and sites; (**B**) volcano map of iWAT Kla-upregulated modification sites; (**C**) heatmap of iWAT Kla-upregulated proteins; and (**D**) the number of Kla sites for each iWAT-upregulated protein.

**Figure 7 metabolites-13-00647-f007:**
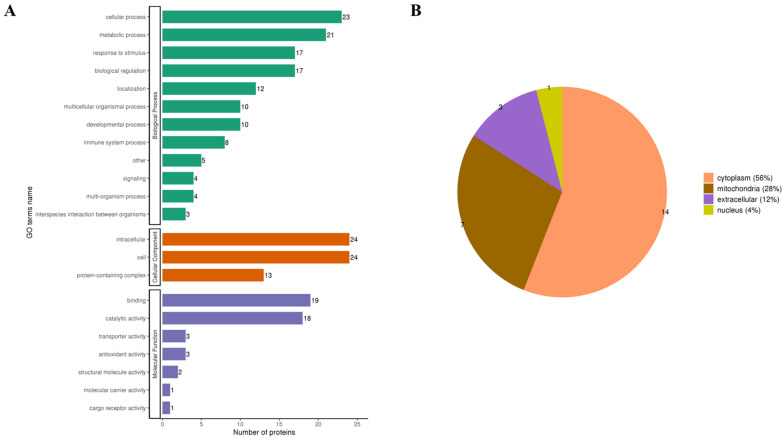
GO and subcellular localization classification of differentially Kla-upregulated proteins. (**A**) iWAT Kla-upregulated protein GO annotation classification; (**B**) iWAT Kla-upregulated protein subcellular structure annotation classification.

**Figure 8 metabolites-13-00647-f008:**
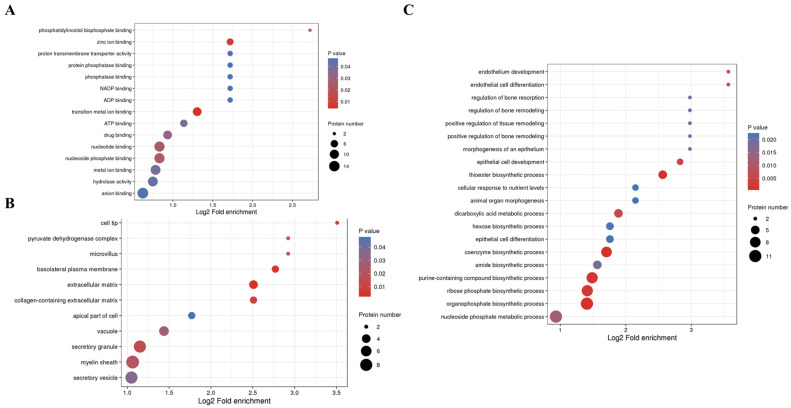
Functional analysis of GO enrichment of iWAT differentially Kla-upregulated protein. (**A**) Molecular function of iWAT Kla-upregulated protein enrichment; (**B**) cellular component of iWAT Kla-upregulated protein enrichment; and (**C**) biological processes of iWAT Kla-upregulated protein enrichment.

**Figure 9 metabolites-13-00647-f009:**
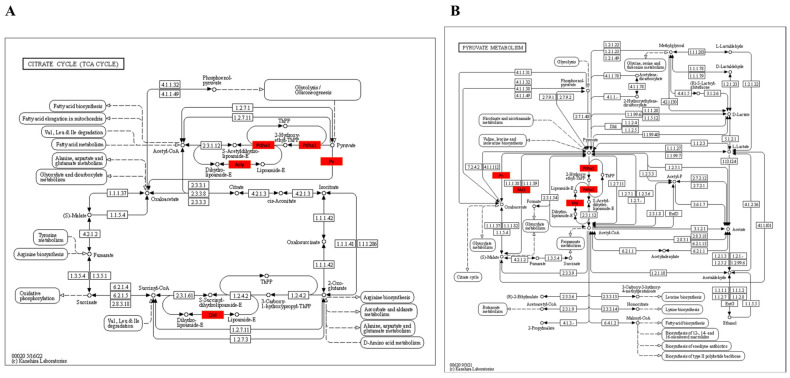
Functional analysis of GO enrichment of iWAT differentially Kla-upregulated protein. (**A**) iWAT Kla-upregulated protein enriched citrate cycle pathway; (**B**) iWAT Kla-upregulated protein enriched pyruvate metabolic pathway; (**C**) iWAT Kla-upregulated protein enriched glycolysis/gluconeogenesis pathway; and (**D**) iWAT Kla-upregulated protein enriched oxidative phosphorylation pathway. The names of the proteins in the red squares are Kla up-regulated proteins. The original KEGG pathway diagram can be viewed at www.kegg.jp.

## Data Availability

The original data of this study are available from the corresponding author upon reasonable request. The data are not publicly available due to being involved in another unpublished study.

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
