# Peer review of "High-Intensity Interval Training Induces Protein Lactylation in Different Tissues of Mice with Specificity and Time Dependence"

_metabolites, 2023, doi:10.3390/metabo13050647_

Round 1

Reviewer 1 Report

The manuscript High-intensity interval training specifically and time-de- 2 pendently induces protein lactylation of different tissues in 3 mice” by Huang et al., and coordinated by Dr. Jing Zhang is an interesting study on protein panlactylation level in HIIT mice. This manuscript is well written and of much interest to the Journal’s Readers. We have concerns about the interpretation of protein lactylation in body tissues. Thus, we do not understand whether protein lactylation is a regular chemical reaction of excess of lactic acid, which react with proteins at random, or this kind of reactions h a special function. We also wonder what effect may have lactic acid in cancerous tissues in the body related to this amazing reaction.  

Author Response

Dear Reviewer:

My colleagues and I thank you and the referees for your effort and time in reviewing our manuscript again and very much appreciate your constructive suggestions.

Sincerely yours,

Wenhua Huang, PhD

Reviewer 2 Report

It is a really well-designed and written paper. Also, it is very interesting and valuable.

I have only some minor comments:

1. Please ensure that the number of words in the abstract is no more than 250.

2. please add the limitations of the study and future perspectives in the discussion section (last paragraph).

3. Supplementary Materials, Appendix A  please delete because no data was included.

4. please supplement the data availability statement (line 555).

5.  I recommend adding the graphical abstract.

Author Response

(The authors gave the same response as above.)

Reviewer 3 Report

The study investigates the impact of a single HIIT session on lactylation of proteins in mouse tissues.  There is great interest in exercise regimes in relation to human health and such studies would attract the attention of clinicians and healthcare  workers in addition to scientific researchers.

 Major comments

There are a number of ambiguities in the design of the study which dilute the significance of the findings. The authors state the HIIT programme to which mice were subjected but do not give their reasons for its design.  Was this with reference to previous literature on mouse training?  It seems that the episode of 1.5 minutes at 85% of Smax is very prolonged compared with human HIIT programmes.  Did the mice approach exhaustion?  If so, the exercise programme was not HIIT. How does this protocol compare with others in the literature?  It is very high intensity but I did not detect the interval involved.

 Figure 1 requires some interpretation.  Did the HIIT mice have very high lactate levels at t=0?  They also seemed to have much higher EE than controls at t=0 (Fig 1D).  How can these findings be explained?  Perhaps the HIIT mice were acclimatized to the running machine but not controls?  If this is the case, then the effects of a single HIIT session were not being measured and the data presented is ambiguous.

 GO and KEGG analyses really contribute very little to this study. It would be more productive to refer to lactylated proteins in terms of what the modifications might do to alter function.  Functional studies are really required to make the current study meaningful.

 The Discussion section is extremely long.  The vast majority of its content is purely speculative and not  based on the findings of the current study.  The Discussion should really be an opportunity to put the current findings in the context of previous work but functional studies are really required to make this feasible.

Minor comments

 The use of English language requires some attention.  The extensive use of the first person plural is not usual in scientific writing.  “Panlactylation”  is not a word that features in mainstream English dictionaries.  There may be a case to be made for a neologism but that would require explanation.  There is some mis-use of vocabulary, for example, in the case of “specifically”

Author Response

(The authors gave the same response as above.)

Round 2

Reviewer 3 Report

The revised version of this manuscript is much improved and I appreciate the effort that the authors have put into clarifying some of their research methods and presentation of results.  I consider the Discussion section to remain overlong and feel it should be condensed to about half its present length.  It is not necessary to describe the metabolism of lactate in terms that would be found in a basic textbook and speculation should be limited to that which draws directly on conclusions of the present work.

Author Response

Dear Reviewer:

My colleagues and I thank you and the referees for your effort and time in reviewing our manuscript again and very much appreciate your constructive suggestions. We revised our manuscript in detail according to the reviewers’ comments, and answered the questions in accompanying letter points to points.

Here, we resubmit our revised manuscript in a marked version that tracks the changes versus our earlier submission, also we provide an unmarked version. 

Sincerely yours,

Wenhua Huang, PhD
